# Ab Initio Molecular Dynamics Investigation of Prenucleation at Liquid–Metal/Oxide Interfaces: An Overview

**Changming Fang** and **Zhongyun Fan** *

Brunel Centre for Advanced Solidification Technology (BCAST), Brunel University London,
Uxbridge UB8 3PH, UK
* Correspondence: zhongyun.fan@brunel.ac.uk

**Abstract:** Prenucleation refers to the phenomenon of atomic ordering in the liquid adjacent to a liquid/solid interface at temperatures above its nucleation temperature. It produces a precursor for heterogeneous nucleation in the liquid and thus has a strong influence on the nucleation process. Oxide particles, including magnesia, spinel, and alumina, are inevitably formed in the liquid during liquid–metal handling and casting. They may act as nucleation sites for potential grain refinement. Knowledge about prenucleation at liquid–metal/oxide (M($l$)/oxide) interfaces is important for an understanding of heterogeneous nucleation during casting. Here, we present an overview of the recent studies on the prenucleation at the M($l$)/oxide interfaces using ab initio molecular dynamics simulation techniques. We observed a wide variety of interfacial chemistry and identified the formation of an ordered metal layer terminating the oxide substrates, such as MgO{1 1 1} (denoting MgO with {1 1 1} surface termination), $\alpha$-Al$_2$O$_3${0 0 0 1}, MgAl$_2$O$_4${1 1 1} and $\gamma$-Al$_2$O$_3${1 1 1} in liquid light metals. The terminating metal atoms are positively charged and form topologically rough layers, which strongly impact the prenucleation at the interfaces. We suggest modification of nucleation potency of the substrate surfaces via elemental segregation to manipulate the solidification processes. This is demonstrated by the segregation of La atoms at the Al($l$)/$\gamma$-Al$_2$O$_3$ interfaces.

**Keywords:** liquid–metal/oxide interfaces; prenucleation; ab initio molecular dynamics modeling; heterogeneous nucleation; impurity segregation; solidification

## 1. Introduction

The densities of the light metals, Mg and Al, which have close-packed structures [1,2], are about one-fifth and one-third of that of iron, respectively. Moreover, Mg and Al metals have unique properties, such as high specific strengths and excellent castability. Thus, Mg and Al alloys have been widely used in a variety of engineering fields, particularly in the automotive and aerospace industries [3,4]. The current regulations on environmental protection demand further improvements in the chemical and mechanical properties of light alloys. One important approach is to achieve solidified products of fine and uniform microstructures by controlling the solidification processes [5–7].

The conventional way to obtain a fine and uniform microstructure of cast alloys is via the addition of grain refiners during casting [4,7]. The added chemicals remain in the cast parts, which may hinder the recycling of the products. Recently, Fan and co-workers developed a new approach to grain refinement using native oxide particles as nucleation sites [6]. This approach is helpful not only for obtaining cast parts of increased integrity, improved mechanical properties, and reduced casting costs but also for facilitating metal recycling. Meanwhile, the application of this approach for casting alloys of desirable microstructure and properties demands knowledge about the atomic ordering at the interfaces between the liquid metal and the oxide substrate.

There have been both experimental and theoretical efforts to obtain insight into the formation, polymorphs, and morphologies of oxide particles and atomic arrangements at

liquid–metal/oxide (noted as M(*l*)/oxide) interfaces. Here, we provide an overview of the recent advances in the understanding of prenucleation at the M(*l*)/oxide interfaces using ab initio molecular dynamics (AIMD) techniques.

The text of the overview is arranged as follows. In Section 2, we present the background information to lay a foundation for studying the prenucleation at the M(*l*)/oxide interfaces. We first introduce prenucleation and the emerging early-stage solidification framework in Section 2.1; the nature of the native oxide particles in the liquid light metals in Section 2.2; the crystal chemistry of the related oxides and their orientational relationships (ORs) with light metals in Section 2.3; and the recent works on factors influencing the atomic ordering (prenucleation) at the liquid/substrate interface (Section 2.4). Then we present the detailed simulation results and related analysis in Section 3. Based on the simulation results, we propose a new approach to modify the nucleation potency of the oxide substrates using elemental segregation at the liquid/oxide interface, which will be demonstrated in Section 4 by taking La atoms at Al(*l*)/$\gamma$-Al$_2$O$_3$ interfaces as an example. The final summary and perspectives are presented in Section 5.

## 2. Background

### 2.1. Prenucleation and Early Stage Solidification

Solid particles in liquid lower the energy barrier of liquid–solid phase transformation and, thus, facilitate the heterogeneous nucleation processes [5–8]. Heterogeneous nucleation is a widely spreading phenomenon in both science and industrial processes, while homogeneous nucleation (without solid substrates) in liquids is rarely observed.

At temperature above its nucleation temperature, there exists atomic ordering in the liquid metal adjacent to the substrate. This phenomenon is referred to as prenucleation [9–11]. Prenucleation produces a precursor for facilitating the subsequent heterogeneous nucleation, depending on the structural compatibility and chemical interactions between the metal and the substrate [9–14].

Heterogeneous nucleation is a process that creates a two-dimensional (2D) nucleus on a substrate that can template further growth of the solid phase [12,13,15]. Building on the precursor provided by prenucleation, heterogeneous nucleation is deterministic, barrier-less, and completed within a few atomic layers. The detailed mechanism depends on the amplitude of lattice misfit and interfacial interactions.

The next process is grain initiation which is defined as a process that creates a three-dimensional (3D) cap that can grow isothermally at a given undercooling [16]. Grain initiation is governed by the grain initiation criterion [17] and is dependent on nucleant particle size [16,17]. Based on the concept of early-stage of solidification, two distinct grain initiation modes have been identified depending on the interplay between nucleation undercooling ($\Delta T_\text{n}$) and grain initiation undercooling of the largest nucleant particle ($\Delta T_\text{gi}$(1st)): progressive grain initiation (PGI) and explosive grain initiation (EGI). The formed 3D cap in the grain initiation process provides a basis for following free growth [16].

### 2.2. Formation of Oxide Particles in Liquid Light Alloys

Oxide particles, including magnesia [18–22], ($\alpha$- and $\gamma$-) alumina [23–25], and spinel [18,20,26–28], form inevitably in liquid alloys during melting handling and casting. Experimental investigation revealed that magnesia (MgO) particles dominantly exist with its {1 1 1} facets (denoted as MgO{1 1 1}) formed in liquid Mg-alloys [20,21], and alumina ($\alpha$-Al$_2$O$_3$ {0 0 0 1} and $\gamma$-Al$_2$O$_3${1 1 1}) particles in Al melts [23–25]. MgAl$_2$O$_4$ spinel particles with the {1 1 1} facets (MgAl$_2$O$_4${1 1 1}) are formed in Al-Mg based alloy melts [18,20,26–28]. These native oxide particles have nontrivial influences on the mechanical performances of the cast parts. They may also act as potential heterogeneous nucleation sites during solidification. Thus, knowledge about prenucleation at the liquid/oxide interfaces is crucial for controlling the solidification processes for obtaining cast parts of fine and uniform microstructures.

There have been experimental and theoretical efforts to understand the atomic arrangements at the M(*l*)/oxide interfaces. Early experiments were focused on the wetting of single oxide crystals by liquid metals [29,30]. The measured contact angles were used to assess the nucleation potency of the substrates based on the classic nucleation theory [31,32]. The previous work on wetting of ceramics by light metals, including oxide surfaces, was reviewed by Kaplan and co-workers [33]. Various experimental instruments, including electron microscopy techniques, have been applied to investigate the interfacial structures and orientational relationships between the metal and the oxide in the solidified samples, such as the Al(*l*)/α-Al$_2$O$_3$\{0 0 0 1\} interfaces [34–41]. The experiments revealed that liquid atoms adjacent to a solid substrate exhibit density variation in the atomic density profile perpendicular to the substrates, which is referred to as atomic layering.

### 2.3. Crystal Chemistry of the Oxides

Light metal oxides exhibit a wide variety of crystal chemistry. The crystal properties of the native oxides are summarized in Table 1. Both MgO and MgAl$_2$O$_4$ (spinel) have face-centered cubic (FCC) lattices, whereas α-Al$_2$O$_3$ exhibits a rhombohedral lattice [1,42]. γ-Al$_2$O$_3$ also has an FCC lattice and a defective spinel-type structure [43]. The structures of these oxides are schematically shown in Figure 1a–d.

**Table 1.** Crystal properties of MgO, α-Al$_2$O$_3$, γ-Al$_2$O$_3$ and MgAl$_2$O$_4$ spinel.

| Oxides | Lattice, Space Group | Latt. Para. (Å) | Characteristics |
|---|---|---|---|
| MgO | Cub., Fm-3m (Nr.225) | *a* = 4.211 [42] | O are in Mg octahedra |
| α-Al$_2$O$_3$ | Hex., R3c (Nr. 167) | *a* = 4.758 [42] <br> *c* = 12.996 | Each O has 4 Al neighbours <br> Al are in octahedra of O |
| γ-Al$_2$O$_3$ | Cub., Fd-3m (Nr. 227) | *a* = 7.9382 [43] * | Partial Al occupation <br> Each O has 3~4 Al neighbours |
| MgAl$_2$O$_4$ | Cub., Fd-3m (Nr. 227) | *a* = 8.080 [42,44] | Each O has 3 Al and 1 Mg neighbours |

* These data were based on an (averaged) defective spinel model [43].

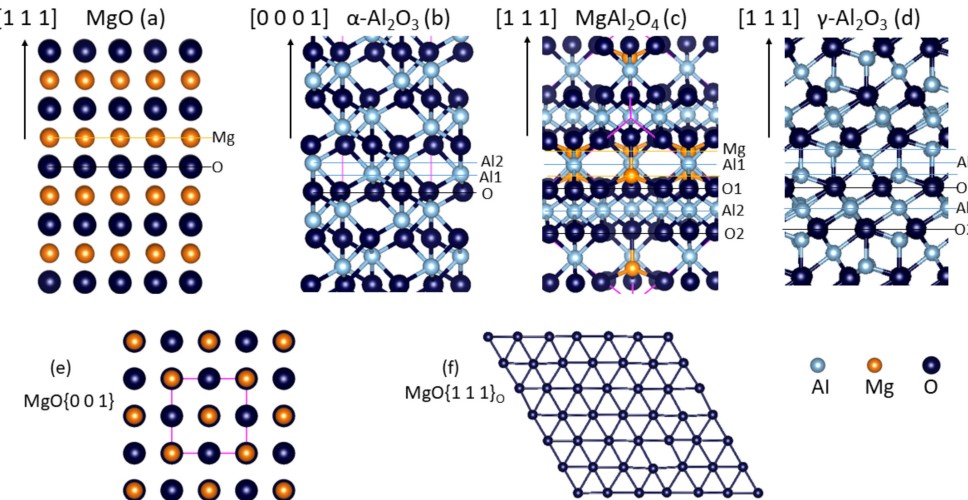

**Figure 1. The schematic structures of oxides are composed of alternatively an O layer and a metal layer as shown in (a–d)**: Schematic structures for (**a**) MgO along its [1 1 1] orientation, (**b**) α-Al$_2$O$_3$ along its [0 0 0 1], (**c**) MgAl$_2$O$_4$ spinel and (**d**) γ-Al$_2$O$_3$ along their [1 1 1] orientation, and related (**e**) MgO\{0 0 1\} and (**f**) MgO\{1 1 1\}$_O$ surface. Selected O and metal (sub)layers as potential substrate surfaces in (**a–d**) are also marked.

MgO has a NaCl-type structure (Figure 1a and Table 1). Both O and Mg form, respectively, face-centered cubic (FCC) sublattices. There is an equal number of Mg and O atoms

at the MgO{0 0 1} surface (Figure 1e), which is the stable surface under ambient conditions. Along the MgO[1 1 1] axis, the structure is composed of alternatively a Mg layer and an O layer (Figure 1a). The atoms at both Mg and O layers form a 2D hexagonal mesh. A cleave along the MgO[1 1 1] produces two smooth surfaces. One is terminated by O and the other by Mg.

$\alpha$-Al$_2$O$_3$ is the ground state of Al$_2$O$_3$ [1,42]. Its structure can be described in a hexagonal lattice. Along its [0 0 0 1] axis, the crystal structure consists of alternatively O- and Al-layers (Figure 1b). The O-layers have a 2D distorted hexagonal lattice. Two-thirds of the octahedral sites between neighboring O layers are occupied by Al, and another one-third are unoccupied. The Al atoms between two O layers form two sublayers. Each Al sublayer consists of a hexagonal sublattice similar to that of graphene. Thus, the Al layer is uneven (Figure 1b). Each Al in $\alpha$-Al$_2$O$_3$ is in a distorted octahedron of O, whereas each O is in a distorted tetragon of Al.

MgAl$_2$O$_4$ belongs to the spinel family with the chemical formula AB$_2$X$_4$; here, A and B are cations, and X is an anion [1,44]. There are 56 atoms in its conventional cell. The structural frame consists of a distorted FCC oxygen sub-lattice (32 O atoms at the Wyckoff sites 32e) which provides 96 interstices. Al atoms occupy half of the 32 octahedral sites (16c), and Mg occupies one-eighth of the 64 tetragonal sites (8a). Each O is coordinated by three Al and one Mg. Along its [1 1 1] axis (Figure 1c), the structure of MgAl$_2$O$_4$ is composed of alternative O layers which have a 2D hexagonal atomic arrangement, an Al layer, or a mixed metal layer (MgAlMg tri-sublayer).

Experiments revealed that $\gamma$-Al$_2$O$_3$ has a defective spinel-like structure with a cubic lattice [43], in which the Al ions are coordinated both tetragonally and octahedrally with O. The Al arrangements in $\gamma$-Al$_2$O$_3$ are based on the replacement of the Mg atoms in spinel (MgAl$_2$O$_4$) by Al. In order to satisfy the charge balance, part of the Al octahedral sites in spinel become unoccupied in $\gamma$-Al$_2$O$_3$. Along the [1 1 1] axis, there are two types of Al layers (Figure 1d). At Al_2, the Al ions occupy two-thirds of the octahedral sites. At Al_1, the Al atoms form three sublayers: a sublayer of octahedrally coordinated Al being sandwiched by two tetragonally coordinated Al sublayers (Figure 1d).

Chemically, the above-mentioned oxides are ionic with Mg$^{2+}$, Al$^{3+}$, and O$^{2-}$ in the ionic model due to the large differences in electronegativity values of the metals (1.61 for Al, 1.31 for Mg in Pauling scale) and the oxygen (3.44). Previous first-principle electronic structure calculations found that both the upper part of the valence bands and the lower part of the conduction bands of these oxides are dominated by O 2p and O 3s characters, respectively [45,46]. This indicates that the structures of these ionic oxides are determined dominantly by the O ion sub-lattices. The role of the metal atoms is to contribute electrons to fill the O 2p orbitals. Therefore, it is rational to consider the dense O layer as the termination for the substrates. Meanwhile, the smooth O-terminated surfaces of these ionic crystals contain net charges, being polar and are, unstable at ambient conditions [47,48]. This can be different when a polar surface is in a condensed metallic environment, whereby the free electrons of metals can compensate for the net charges at the substrate surface [49,50], and thus, the 2D oxygen or metal layers may be used as substrate surfaces in liquid metals.

The lattice misfit between the substrate and metal has been used as a critical factor in heterogeneous nucleation in the metal melt/oxide systems [6,9,15]. Table 2 lists the lattice misfits between the light metals and the oxide substrates.

**Table 2.** Terminating surfaces, orientational relations (ORs) and lattice misfits between metals (M) and the oxide substrates (S), M/S [2]. The lattice misfit is defined as: $f = (d_M - d_S)/d_M \times 100\%$, where $d_S$ represent the lattice spacing of the oxide and $d_M$ of the metal. Data from the literature [20] are included.

| Terminating Surfaces | ORs $\{hkl\}[uvw]_M // \{h_0k_0l_0\}[u_0v_0w_0]_S$ | $d[uvw]_M$ (Å) | $d[u_0v_0w_0]_S$ (Å) | $f(\%)$ |
|---|---|---|---|---|
| Mg{0 0 01}/MgO{1 1 1} | {0001}[1000]M//{111}[100]S | 3.213 [2] | 2.978 [42] | +7.9 |
| Al{1 1 1}/MgO{1 1 1} | {111}[100]M//{111}[100]S | 2.914 [2] | 2.978 [42] | −2.2 |
| Al{1 1 1}/α-Al$_2$O$_3$\{0 0 0 1} | {111} [220]M //{0001}[1000]S | 5.052 * | 4.785 [42] | +5.6 |
| Al{1 1 1}/MgAl$_2$O$_4$\{1 1 1} | {111}[200]M//{111}[100]S | 5.828 (=2 × 2.914) | 5.746 [42] | +1.4 [20] |
| Al{1 1 1}/γ-Al$_2$O$_3$\{1 1 1} | {111}[200]M//{111}[100]S | 5.828 (=2 × 2.914) | 5.631 [43] | +3.3 [20] |

* The new hexagonal cell has a rotation of 30° (R30°) with a = $\sqrt{3}\, a_0$, here $a_0$ is the length of *a*-axis of the conventional hexagonal unit cell [2].

The values of the lattice misfits between the metals and the oxides substrates vary from moderate (1.4%) to large (7.9%), as shown in Table 2. It is expected that the lattice misfit influences the prenucleation at the M(*l*)/oxides interfaces.

There have been both experimental and theoretical efforts to understand the structure and properties of the oxides [1,42–46], their surfaces [47,48], related metal/ceramic joints [51], liquid metal and solid substrate interfaces [37–39,52–54], as well as interfaces between metal and substrates in cast samples [34–36,55–60]. Knowledge about atomic ordering (prenucleation) at liquid/oxide interfaces is important not only for understanding the solidification processes of liquid metals but also for other fields, such as metal-ceramics welding, anticorrosion, and metal-oxide composites.

In brief, chemically, all the light-metal oxides are ionic in nature. Structurally, along the MgO[1 1 1], MgAl$_2$O$_4$[1 1 1], γ-Al$_2$O$_3$[1 1 1], and α-Al$_2$O$_3$[0 0 0 1] axis, the structures are composed alternatively of an O layer and a metal layer in common. The O ions form 2D (distorted) hexagonal sublattices. The metal layers exhibit a variety of atomic arrangements. Moreover, there is a variety of potential lattice matches between the light metals and the oxides substrates. The rich crystal chemistry of the oxide substrates and the variety of lattice misfits between the light metals and oxide substrates indicate the level of complexity of the prenucleation at the M(*l*)/oxide interfaces.

### 2.4. Factors Affecting Prenucleation from Atomic Simulations

Both semiempirical atomistic (atomistic MD) and ab initio molecular dynamics (AIMD) simulation approaches have been applied to investigate atomic arrangements at liquid–metal/solid-substrate interfaces. The studies revealed atomic ordering, including layering and in-plane ordering in the liquid adjacent to a solid substrate. The liquid metal atoms near the substrate exhibit atomic layering along the direction perpendicular to the substrate, whereas the liquid atoms further away from the substrate display disordering, having a liquid-like nature. There are six recognizable atomic layers at the Al(*l*)/Al(*s*) interface, with the peak density decreasing with increasing distance from the interface [9–11]. The liquid metal atoms in the individual layers also exhibit atomic ordering with similarity to those in a solid. The in-plane ordering coefficient, $S(z)$ is only significant in the first three atomic layers and decreases with the distance from the substrate. The in-plane ordering in the fourth and subsequent layers is minor.

In the last few years, much effort has been made to understand the factors affecting prenucleation at liquid/substrate interfaces. At present, four important factors, temperature, lattice misfit, the chemistry of the substrate, and atomic surface roughness of the substrate, have been investigated. These factors and their impacts are summarized in Table 3.

Using an AIMD simulation technique, Fang et al. investigated the temperature effect on prenucleation at a generic liquid–metal/solid system with the substrate atoms being pinned [10]. The simulations revealed that even at 2000 K, there is a certain degree of atomic layering in the liquid metal adjacent to the pinned substrate with two recognizable layers. However, the in-plane ordering coefficient at the interface at 2000 K is minor. Both layering and in-plane ordering in the liquid adjacent to the substrate increase with decreasing temperature. The number of recognizable layers of liquid metal increases from three at 2000 K to six at 950 K. The in-plane ordering coefficient of the first liquid layer increases from 0.01 at 2000 K to 0.54 at 950 K. Similar behavior was observed in the semiempirical atomistic MD study at the liquid/solid interface [9,12].

Men and Fan performed atomistic MD simulations on the effects of the structural factor on the atomic ordering in generic liquid metal/solid systems with varying lattice misfit [9,11–13]. They revealed that lattice misfit strongly affects the in-plane atomic ordering in the liquid adjacent to the substrate. However, it has little influence on the layering of liquid atoms adjacent to the substrates.

AIMD simulations were performed to obtain insight into the effect of substrate chemistry on prenucleation at liquid-Al/solid-M (Al($l$)/M($s$) in short, M= Al, Ag, W, and Cd) interfaces [10]. The chosen substrate metals have small lattice misfits (<1%) with Al. They have different heats of mixing with Al [61], which was used as a measure of the interfacial chemical interaction. The AIMD simulations revealed a trend of substrate chemistry in the prenucleation at the interfaces. For a substrate that is chemically affinitive to metal, both layering and in-plane ordering at the interface are enhanced, whereas, for a chemically repulsive substrate, the prenucleation at the interface is reduced.

**Table 3.** Factors affecting the prenucleation at liquid–metal/solid-metal interfaces.

| Factors | Definition | Effects on Prenucleation |
|---|---|---|
| Temperature [10] | $T$ | Prenucleation increases with decreasing $T$. |
| Lattice misfit [9,11] | $f = (d_m - d_s)/d_m \times 100\%$ | $f$ hinders in-plane ordering but hardly on layering. |
| Substrate chemistry [10] | $\Delta H_{\mathrm{mix}}$ | Affinitive substrates promote prenucleation, whereas repulsive substrates do oppositely. |
| Atomic roughness [14,62] | $R = [\sum(|\Delta z(i)|/d_0)]/N_z$ | $R$ deteriorates both in-plane-ordering and layering. |

Jiang et al. systematically investigated the effects of atomic surface roughness of the substrates on prenucleation using atomistic molecular dynamics (MD) simulations [14]. They revealed strong impacts of atomic surface roughness on the prenucleation. Atomic roughness deteriorates both atomic layering and in-plane ordering in the liquid at the interfaces. Moreover, for an amorphous substrate with a rough surface, the liquid atoms adjacent to the substrate behave liquid-like, and there is little prenucleation at the interface [11,14].

## 3. AIMD Investigations of Prenucleation at the M($l$)/Oxide Interfaces

### 3.1. Supercells and Details of AIMD Simulations

High-resolution transmission electron microscopy (HR-TEM) investigations revealed that the native oxide particles have dominant MgO{1 1 1}, $\gamma$-Al$_2$O$_3${1 1 1}, MgAl$_2$O$_4${1 1 1}, and $\alpha$-Al$_2$O$_3${0 0 01} facets in liquid metals and their alloys. Based on the experimental results, we built supercells for the M($l$)/oxide interfaces for the AIMD simulations. The built supercells are hexagonal. The dimension of the $a$-axis is $a = (5\sqrt{2}/2) a_0$ for MgO{1 1 1}, $a = 3 a_0$ for $\alpha$-Al$_2$O$_3${0 0 0 1} and $a = (3\sqrt{2}/2) a_0$ for MgAl$_2$O$_4$ and $\gamma$-Al$_2$O$_3$ spinel ($a_0$ is the lengths of the $a$-axis of the oxides with consideration of the thermal expansion at the simulation temperature). The dimension of the $c$-axis is determined by the oxide slab and the number of metal atoms with the atomic volume at the simulation temperature [2]. Analysis of the structures of the oxides in Figure 1 and their ORs with metals in Table 2 reveals the possible surfaces of the oxide substrates in liquid metals (Table 4). We choose independent interfaces for each system

(Table 4). To study the evolutions of atomic ordering during the simulations and to assess the convergences of simulations, we also choose some dependent interfaces that are given in Table 4.

**Table 4.** Inputs of the designed M(*l*)/oxide interfaces for AIMD simulations. All cells are hexagonal.

| Systems | Possible Interfaces | | Cells' Paras. (Å) | | Number of Atoms | | |
|---|---|---|---|---|---|---|---|
| | | | a | c | Mg | Al | O |
| Mg(*l*)/MgO{111} [50] | (i). | Mg(*l*)/MgO{111}$_{Mg}$ | 14.90 | 64.62 | 425 | - | 100 |
| | (ii). | Mg(*l*)/MgO{111}$_{O}$ | 14.90 | 64.62 | 425 | - | 100 |
| Al(*l*)/MgO{111} [49,63] | (i). | Al(l)/MgO{111}$_{Mg}$ | 14.90 | 48.75 | 125 | 425 | 100 |
| | (ii). | Al(l)/MgO{111}$_{O}$ | - | - | - | - | - |
| | (iii). | Al(l)/MgO{111}$_{Al}$ | 14.90 | 48.50 | 75 | 450 | 100 |
| Al(*l*)/α-Al$_2$O$_3$ {49} | (i). | Al(*l*)/α-Al$_2$O$_3${0001}$_{O}$ | 14.40 | 51.82 | - | 524 | 81 |
| | (ii). | Al(*l*)/α-Al$_2$O$_3${0001}$_{Al\_1}$ | 14.40 | 55.31 | - | 558 | 81 |
| | (iii). | Al(*l*)/α-Al$_2$O$_3${0001}$_{Al\_2}$ | 14.40 | 53.57 | - | 541 | 81 |
| | (iv). | Al(*l*)/α-Al$_2$O$_3${0001}$_{Al\_2}$ | 14.40 | 40.21 | - | 360 | 108 |
| Al(*l*)/MgAl$_2$O$_4${111} [64] | (i). | Al(*l*)/MgAl$_2$O$_4${111}$_{O\_1}$ | 17.24 | 31.51 | 18 | 387 | 144 |
| | (ii). | Al(*l*)/MgAl$_2$O$_4${111}$_{O\_2}$ | 17.24 | 31.72 | 36 | 369 | 144 |
| | (iii). | Al(*l*)/MgAl$_2$O$_4${111}$_{(O\_2)Al\_2}$ | 17.24 | 42.62 | 36 | 549 | 144 |
| | (iv). | Al(*l*)/MgAl$_2$O$_4${111}$_{(O\_1)Mg}$ | 17.24 | 32.13 | 36 | 387 | 144 |
| | (v). | Al(*l*)/MgAl$_2$O$_4${111}$_{(O\_1)MgAl}$ | - | - | - | - | - |
| | (vi). | Al(*l*)/MgAl$_2$O$_4${111}$_{(O\_1)MgAlMg}$ | 17.24 | 43.19 | 54 | 531 | 144 |
| Al(*l*)/γ-Al$_2$O$_3${111} [62] | (i). | Al(*l*)/γ-Al$_2$O$_3$ {111}$_{O\_1}$ | 17.06 | 40.58 | - | 522 | 144 |
| | (ii). | Al(*l*)/γ-Al$_2$O$_3${111}$_{O\_2}$ | 17.06 | 40.58 | - | 522 | 144 |
| | (iii). | Al(*l*)/γ-Al$_2$O$_3$ {111}$_{(O\_2)Al}$ | - | - | - | - | - |
| | (iv). | Al(*l*)/γ-Al$_2$O$_3$ {111}$_{(O\_1)Al}$ | - | - | - | - | - |
| | (v). | Al(*l*)/γ-Al$_2$O$_3$ {111}$_{(O\_1)AlAl}$ | - | - | - | - | - |
| | (vi). | Al(*l*)/γ-Al$_2$O$_3$ {111}$_{(O\_1)AlAlAl}$ | - | - | - | - | - |

A pseudo-potential plane-wave approach based on the density-functional theory (DFT) was used for the present study. This approach was implanted into the first-principles code VASP (Vienna ab initio simulation package) [65]. VASP permits variable fractional occupation numbers, working well for insulating/metallic interfaces [65,66]. The molecular dynamics simulation uses the finite-temperature density functional theory of one-electron states, the exact energy minimization and calculation of the exact Hellmann-Feynman forces after each MD step using the preconditioned conjugate techniques, and the Nosé dynamics for generating a canonical NVT ensemble. The Gaussian smearing was employed with the width of smearing (0.1 eV). The code also utilizes the projector augmented-wave (PAW) method [67] within the generalized gradient approximation [68]. The electronic configurations used are Mg ([Ne] 3s$^2$ 3p$^0$), Al ([Ne] 3s$^2$ 3p$^1$) and O ([He] 2s$^2$ 2p$^4$).

For electronic structure calculations, we used cut-off energies of 400.0 eV for the wave functions and 550.0 eV for the augmentation functions. Reasonably dense *k*-meshes were used for sampling the electronic wave functions, e.g., a $2 \times 2 \times 1$ (8 *k*-points) in the Brillouin zone (BZ) of the supercell of the interfaces, based on the Monkhorst-Pack scheme [69]. For the AIMD simulations of the interfaces, we employed cut-off energy of 320 eV and the Γ-point in the BZ, considering the lack of periodicity of the whole system in molecule/solid-substrate and liquid/solid interfaces [66,70–73]. Test simulations using different cut-off energies demonstrated that the settings are reasonable.

We prepared liquid Al or Mg samples by equilibrating at 3000 K for 2000 steps (1.5 fs per step). Then the obtained liquid was cooled to the desired temperature. We used the obtained liquid Al or Mg samples together with the oxide substrates for building the M(*l*)/oxide simulation systems. We employed two different approaches in the AIMD simulation depending on the systems. Full relaxation of atoms is used all the time. The other

adopted a two-step approach: We first performed AIMD simulations with the substrate O atoms pinned for about 2 ps (1.5 ps per step). Then, we equilibrated the systems further with full relaxation of the substrate atoms for another 4000 to 7000 steps. The time-averaged method was used to sample the interfaces over 3.0 to 4.5 ps to ensure statistically meaningful results [66,70].

Next, we discuss the prenucleation at the Al($l$)/Al$_2$O$_3$ interfaces and at the interfaces between liquid metals (Mg, Al) and MgO{1 1 1} separately.

### 3.2. Prenucleation at the Al(l)/Al$_2$O$_3$ Interfaces

We first discuss the atomic evolutions at the M($l$)/oxide interfaces using the Al($l$)/Al$_2$O$_3$ interfaces at 1000 K as an example. During simulations, we observed that the liquid Al atoms move quickly to the oxide substrates, forming a terminating layer. The number of atoms in this newly formed metal layer becomes gradually stabilized, forming a new substrate surface within 2 ps. Then, the atoms/ions in the oxide substrates vibrate around their equilibrium positions, while the liquid metal atoms away from the interfaces move around freely. The simulations revealed that the behavior of liquid atoms near the substrates varies depending on the nature of the substrate surfaces.

The rate for reaching equilibration of the M($l$)/oxide interfaces can also be observed from the variations of total valence-electrons free energies with the simulation time, as sampled in Figure 2 for the Al($l$)/Al$_2$O$_3$ interfaces [49,62]. The energy reduction is fast in the first 0.5ps and then levels off with time. The time required for reaching a constant energy value varies for the different interfaces. As shown in Figure 2, it takes a longer time to reach equilibrium (~2 ps) at Al($l$)/$\gamma$-Al$_2$O$_3$\{0 0 0 1\}$_{Al2}$ as compared with that at the Al($l$)/$\alpha$-Al$_2$O$_3$ interface (~1.5 ps). The AIMD simulations showed that the interfaces reach equilibrium after 2 ps overall. Similar evolution behavior of atomic arrangements was observed during the simulations of other M($l$)/oxide interfaces.

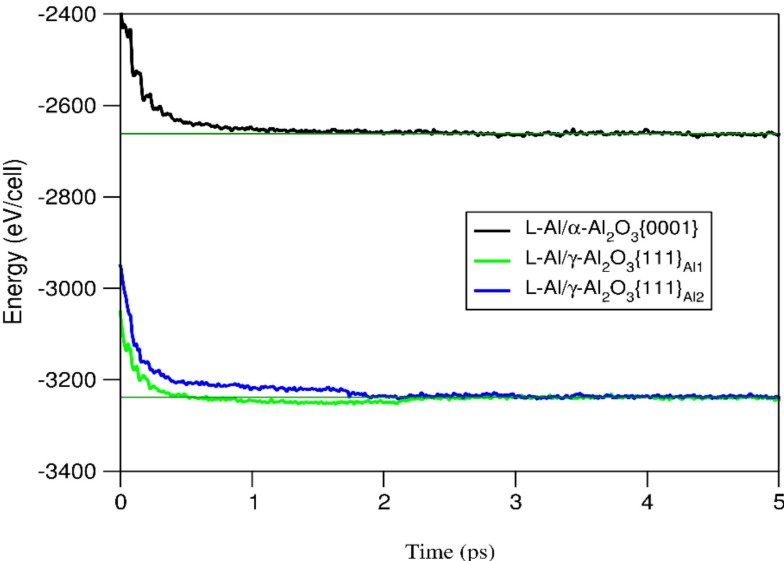

**Figure 2.** The two Al(l)/$\gamma$-Al$_2$O$_3$\{1 1 1\} interfaces have similar energies at equilibrium at 1000 K. Dependences of total valence-electrons energies of the Al(l)/Al$_2$O$_3$ systems on simulation time [49,62]. The straight dark-green lines represent the energies at equilibrium.

We present the snapshots of the equilibrated Al($l$)/Al$_2$O$_3$ interfaces at 1000 K in Figure 3. The related atomic density profiles for configurations over 3 ps are shown in Figure 4a. The in-plane ordering coefficients for the time-averaged atomic positions over 3 ps are plotted in Figure 4b.

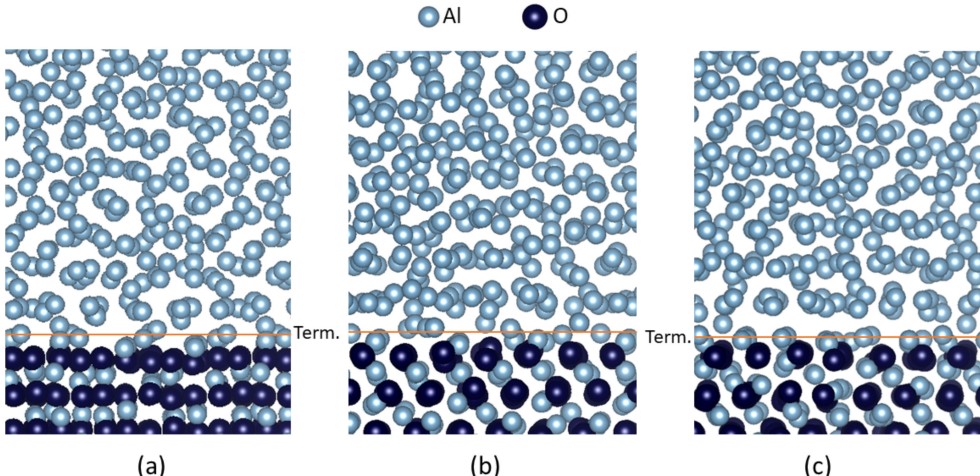

**Figure 3. An Al layer terminates the oxide substrates in the liquid.** Snapshots of the equilibrated Al($l$)/$\alpha$-Al$_2$O$_3$\{0 0 0 1\} (**a**), Al($l$)/$\gamma$-Al$_2$O$_3$\{1 1 1\}$_{Al\_1}$ (**b**) and Al($l$)/$\gamma$-Al$_2$O$_3$\{1 1 1\}$_{Al\_2}$ (**c**) systems [49,62]. The terminating Al layers are marked by the orange lines.

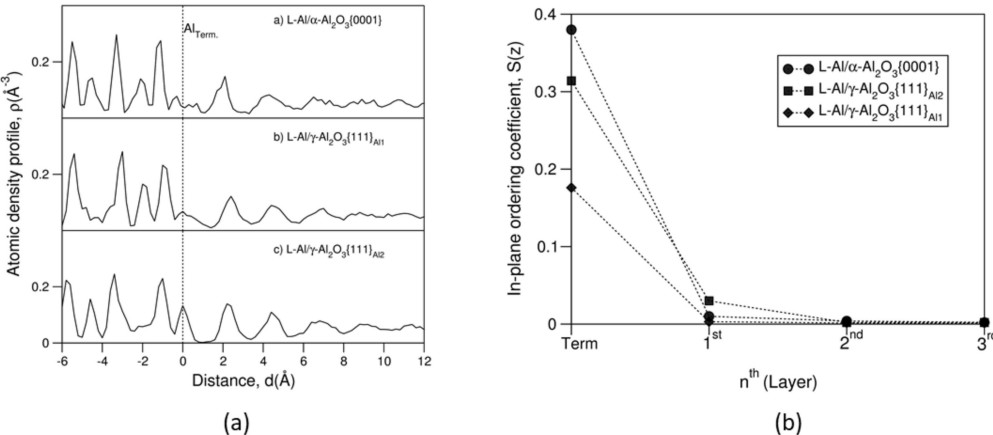

**Figure 4. The Al terminating layers can be either flat or rough and atomic ordering at the Al($l$)/Al$_2$O$_3$ interfaces is overall weak.** Atomic density profiles (**a**) and in-plane ordering coefficient (**b**) at the equilibrated Al($l$)/Al$_2$O$_3$ interfaces [49,62]. The vertical dotted line in 4a marks the terminating Al layers.

The terminating Al layers at the Al($l$)/$\gamma$-Al$_2$O$_3$\{1 1 1\}$_{Al\_2}$ interface are flat and contain vacancies. The terminating Al layers at the Al($l$)/$\gamma$-Al$_2$O$_3$\{1 1 1\}$_{Al\_1}$ and Al($l$)/$\alpha$-Al$_2$O$_3$\{0 0 0 1\} interfaces contain both vacancies and displacements along the direction perpendicular to the substrates (Figure 4a). Analysis showed that there is moderate atomic ordering in the first Al layer but little in the second Al layer in the Al($l$)/$\gamma$-Al$_2$O$_3$\{1 1 1\}$_{Al\_2}$ interface (Figure 4b). At the Al($l$)/$\gamma$-Al$_2$O$_3$\{1 1 1\}$_{Al\_1}$ and Al($l$)/$\alpha$-Al$_2$O$_3$\{0 0 0 1\} interfaces even the first Al layer has little atomic ordering.

The epitaxial nucleation model indicates that the substrate surface atoms template ordering in the nearby liquid to nucleate [74]. Here, time-averaged atomic arrangements at the Al layers at the Al($l$)/Al$_2$O$_3$ interfaces are shown in Figure 5. The terminating Al atoms at the Al($l$)/$\gamma$-Al$_2$O$_3$\{1 1 1\} interfaces show stronger localization than those at the Al($l$)/$\alpha$-Al$_2$O$_3$\{0 0 0 1\}. At the first layer of the Al($l$)/$\alpha$-Al$_2$O$_3$\{0 0 0 1\} and Al($l$)/$\gamma$-Al$_2$O$_3$\{1 1 1\}$_{Al\_1}$ interfaces, Al atoms are more liquid-like as compared with those at the Al($l$)/$\gamma$-Al$_2$O$_3$\{1 1 1\}$_{Al\_2}$ interface. Thus, prenucleation at the Al($l$)/$\gamma$-Al$_2$O$_3$\{1 1 1\}$_{Al\_2}$ interface is more pronounced than at the other interfaces.

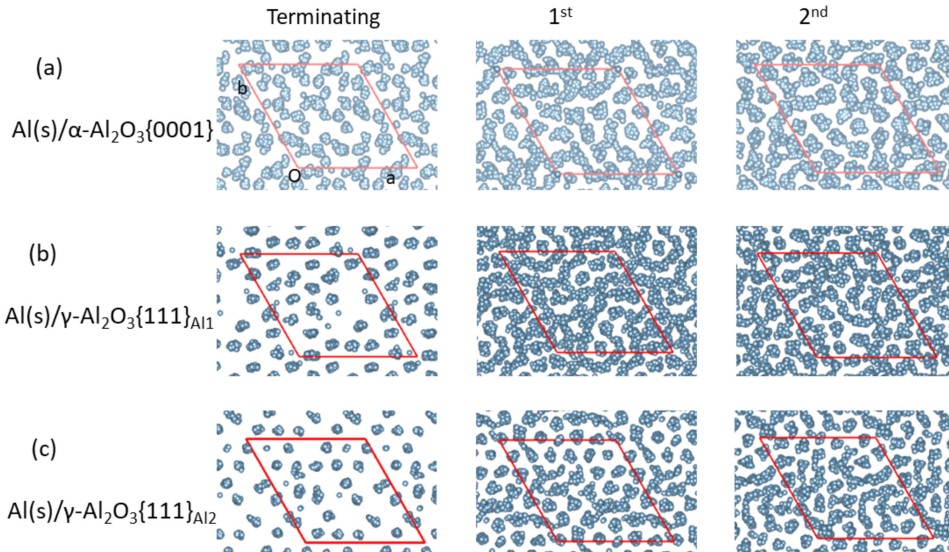

**Figure 5.** Al(*l*)/γ-Al$_2$O$_3${1 1 1}$_{Al\_2}$ interface exhibits more notable prenucleation with atomic ordering and stronger localization at the 1st Al layer than the other interfaces. Time averaged atomic arrangements in the terminating, 1st and 2nd Al layers at the equilibrated (**a**) Al(*l*)/α-Al$_2$O$_3${0 0 0 1}, (**b**) Al(*l*)/γ-Al$_2$O$_3${1 1 1}$_{Al\_1}$ and (**c**) Al(*l*)/γ-Al$_2$O$_3${1 1 1}$_{Al\_2}$ interfaces [49,62].

There are apparent holes at the termination layers of the Al(*l*)/γ-Al$_2$O$_3${1 1 1} systems (Figure 5b,c). The occupation of the octahedral sites in the terminating layers was analyzed using the time-averaged atomic positions over 3 ps: The occupation rate at the terminating layer by Al is 54.0% in the Al(*l*)/γ-Al$_2$O$_3${1 1 1}$_{Al\_1}$ system and 58.1% in the Al(*l*)/γ-Al$_2$O$_3${1 1 1}$_{Al\_2}$ system. These values are lower than that in the bulk alumina (66.7%), but they are comparable with that for the Al(*l*)/α-Al$_2$O$_3${1 1 1} system (55.9%). These occupation rates of the terminating Al layers are notably lower than those in the Al(*l*)/MgAl$_2$O$_4${1 1 1} system (70.4 to 75.0%) [64]. This reflects the role of the crystal chemistry of the substrates. The simulations also revealed structural coupling of the terminating Al layer to the γ-Al$_2$O$_3${1 1 1} substrate: In the Al(*l*)/γ-Al$_2$O$_3${1 1 1}$_{Al\_1}$ system, the multiple-peaked terminating Al layer is coupled to the single peak in the subsurface Al layer, whereas in the Al(*l*)/γ-Al$_2$O$_3${1 1 1}$_{Al\_2}$ system the single-peaked Al terminating layer is accompanied by the multiple-peaked subsurface Al layer in the substrate.

In summary, we have the following conclusions:

(i).     The Al and O atoms/ions in the substrates form layers of atomic ordering;
(ii).    The substrates are terminated by a layer of Al atoms. The terminating Al atoms form a single peak at the Al(*l*)/γ-Al$_2$O$_3${1 1 1}$_{Al\_2}$ interface, whereas they form multiple peaks (three sublayers) at the Al(*l*)/γ-Al$_2$O$_3${1 1 1}$_{Al\_1}$ interface;
(iii).   The Al-O interatomic distances between the terminating Al atoms and the outmost O ions are close to those in the bulk substrate. This indicates that the terminating Al atoms are chemically bonded to the substrates, becoming an integrated part of the substrates;
(iv).    Both layering and in-plane ordering at the Al(*l*)/γ-Al$_2$O$_3${1 1 1}$_{Al\_2}$ interface are more pronounced than that at Al(*l*)/γ-Al$_2$O$_3${1 1 1}$_{Al\_1}$ and Al(*l*)/α-Al$_2$O$_3${0 0 0 1}.
(v).     There are atomic vacancies in the terminating Al layers.

Charge and charge transfer provide direct evidence about chemical interactions in a compound or at an interface. We performed electronic structure calculations of the equilibrated Al(*l*)/Al$_2$O$_3$ interfaces. Via Bader's model [75,76], the charges at the atomic sites at the interfaces based on the calculated electron density distributions were obtained and are shown in Figure 6.

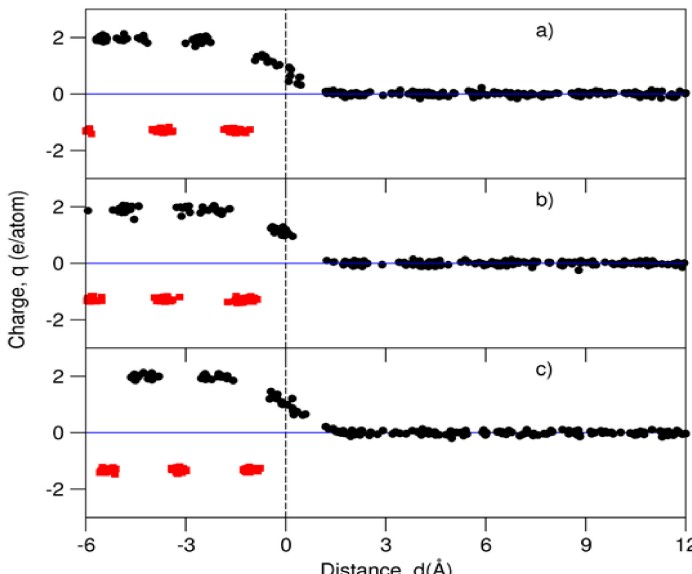

**Figure 6. There is charge transfer occurring from the terminating Al atoms to the outmost O ions.** Charges at the atomic sites across the Al($l$)/$\gamma$-Al$_2$O$_3$\{1 1 1\}$_{Al\_1}$ (**a**), Al($l$)/$\gamma$-Al$_2$O$_3$\{1 1 1\}$_{Al\_2}$ (**b**) and Al($l$)/$\alpha$-Al$_2$O$_3$\{0 0 0 1\} (**c**) interfaces [49,62]. The red squares represent charges at O, black at Al.

The O and Al ions in the substrates are charged with $-1.3$e and $+2.0$e, respectively (Figure 6). The Al away from the substrates is electronically neutral. The terminating Al atoms are positively charged with electrons transferred to the outmost O ions. Thus, chemically the terminating Al ions/atoms are bonded to the substrates, being the integrated part of the substrates.

In summary, the AIMD simulations revealed an Al layer terminating the Al($l$)/Al$_2$O$_3$ substrate. The terminating Al atoms are positively charged, chemically bonded to the outmost O ions, thus being an integral part of the substrates. The terminating Al layers contain vacancies and displacements vertical to the substrate surface, resulting in atomic roughness. Consequently, the prenucleation at the Al($l$)/Al$_2$O$_3$ interfaces is overall weakened, and they are impotent substrates for heterogeneous nucleation of solid Al.

### 3.3. Prenucleation at the M(l)/MgO{1 1 1} (M = Mg, Al) Interfaces

The AIMD simulations for the MgO\{1 1 1\} substrates in liquid Mg and Al [49,50,63]. Snapshots of the equilibrated interfaces are shown in Figure 7. The related atomic density profiles are shown in Figure 8.

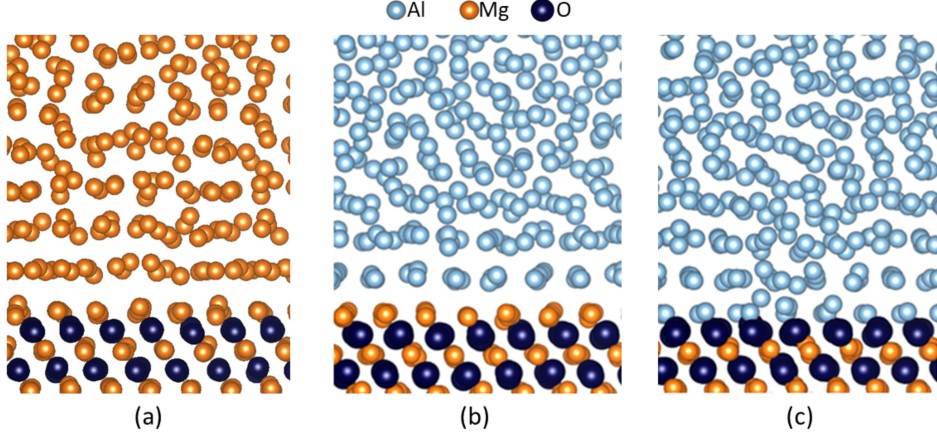

**Figure 7. A metal layer terminates the MgO{1 1 1} substrates in the liquid.** Snapshots of the equilibrated Mg($l$)/MgO\{1 1 1\} (**a**), Al($l$)/MgO\{1 1 1\}$_{Mg}$ (**b**) and Al($l$)/MgO\{1 1 1\}$_{Al}$ (**c**) systems [49,50,63].

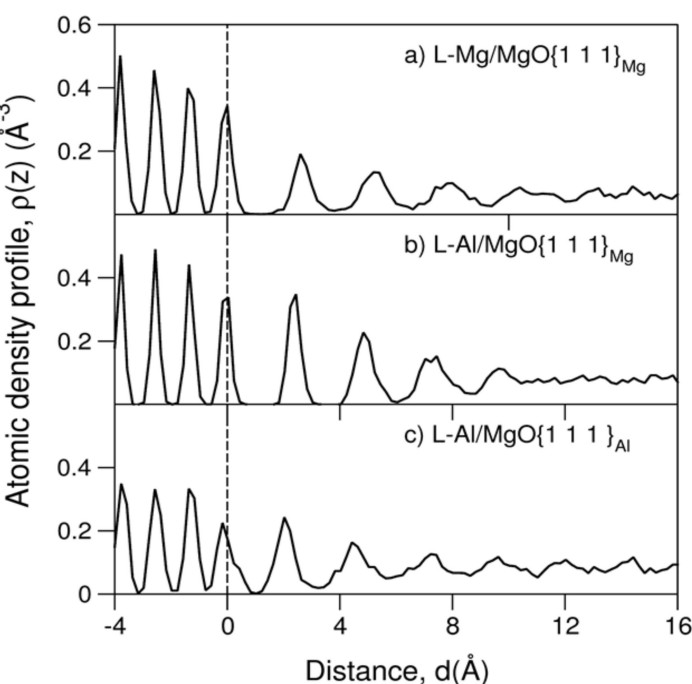

**Figure 8.** The Al(*l*)/MgO{1 1 1}$_{Mg}$ interface exhibits more pronounced prenucleation. Atomic density profile for (**a**) Mg(*l*)/MgO{1 1 1}, (**b**) Al(*l*)/MgO{1 1 1}$_{Mg}$ and (**c**) Al(*l*)/MgO{1 1 1}$_{Al}$ interfaces [49,50,63]. The vertical dotted line represents the terminating metal layer.

The Mg and O layers in the MgO{1 1 1} substrates are well-ordered and solid-like. This corresponds to the strong chemical bonding in MgO. Although the liquid metal atoms away from the substrates display disordering and behave liquid-like, those near the interfaces exhibit layering (Figures 7 and 8). There is a metal layer terminating the MgO{1 1 1} substrates. The terminating Mg atoms at the M(*l*)/MgO{1 1 1}$_{Mg}$ (M= Mg, Al, Figure 7a,b, and Figure 8a,b) form a single peak, whereas the terminating Al atoms at Al(*l*)/MgO{1 1 1}$_{Al}$ form a peak and a shoulder at 0.8Å away from the artificial dividing interface (Figures 7c and 8c). There is an isolated Al layer (the first metal layer) at the Al(*l*)/MgO{1 1 1}$_{Mg}$ interface, whereas the first Al layer at the Al(*l*)/MgO{1 1 1}$_{Al}$ interface is admixed with the second Al layers.

We also analyzed the time-averaged atomic arrangements for the liquid atoms adjacent to the substrates at the M(*l*)/MgO{1 1 1} interfaces, as shown in Figure 9. The related in-plane ordering coefficients are plotted in Figure 10.

In the Al(*l*)/MgO{1 1 1}$_{Mg}$ system, the terminating Mg layer at the Al(*l*)/MgO{1 1 1}$_{Mg}$ interface has a sharp peak in the atomic density profile (Figure 8b), and the Al atoms in the first layer display significant ordering (Figure 9). Similarly, the terminating Mg atoms at the Mg(*l*)/MgO{1 1 1} interface form a sharp peak in the density profile (Figure 8a) and display strong localization (Figure 9). Meanwhile, there are some unoccupied sites at the terminating layer. Statistics provided the site occupation of 92% [50]. Comparatively, the terminating Al atoms at the Al(*l*)/MgO{1 1 1}$_{Al}$ interface are less localized and contain some disordering (Figure 9). This is reflected in the in-plane ordering coefficients (Figure 10). The in-plane ordering coefficients of the terminating Mg layers at both Mg(*l*)/MgO{1 1 1} and Al(*l*)/MgO{1 1 1}$_{Mg}$ interfaces are close to around 0.6, being significantly larger than that (about 0.2) at the Al(*l*)/MgO{1 1 1}Al interface.

Figures 9 and 10 also show that the first Al atoms at the Al(*l*)/MgO{1 1 1}$_{Mg}$ interface show localized, solid-like behaviors (Figure 9b), whereas the first Mg (Figure 9a) and Al atoms (Figure 9c) are more liquid-like. Correspondingly, the in-plane ordering coefficient of the first Al layer at the Al(*l*)/MgO{1 1 1}$_{Mg}$ interface is about 0.33, whereas the coefficients of the first metal layers at the other interfaces are very low (<0.03) as shown in Figure 10. The latter comes from either the atomic vacancies at the flat Mg layer terminating

the MgO{1 1 1) (Figure 9a) or the atomic roughness caused by the displacement at the terminating Al layer at the Al(*l*)/MgO{1 1 1}$_{Al}$ interface.

The present study reveals that the atomic structure and the interfacial interactions are crucial in the determination of the prenucleation at a liquid/solid interface. This also indicates possibilities to control the prenucleation or nucleation potency of the substrates via, e.g., segregating impurity atoms at the interfaces.

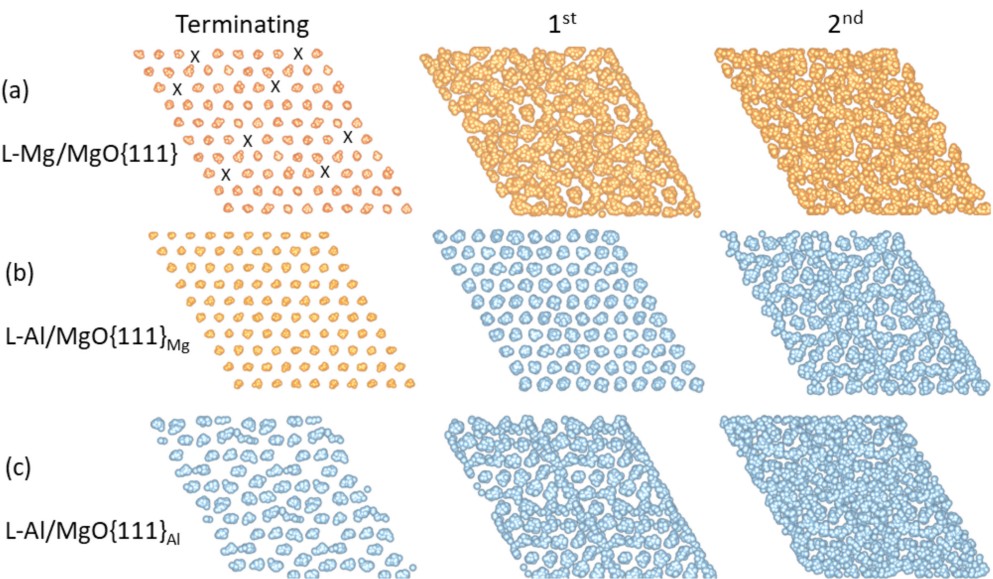

**Figure 9.** **There are atomic vacancies (marked by X) at the terminating layer at the Mg(*l*)/MgO{1 1 1} and the termination metal atoms display localized nature.** Time averaged atomic arrangements in the terminating, 1st and 2nd Al layers at the (**a**) the equilibrated Mg(*l*)/MgO{1 1 1}, (**b**) Al(*l*)/MgO{1 1 1}$_{Mg}$ and (**c**) Mg(*l*)/MgO{1 1 1}$_{Al}$ interfaces [49,50,63]. The meaning of the spheres is the same as in Figure 7.

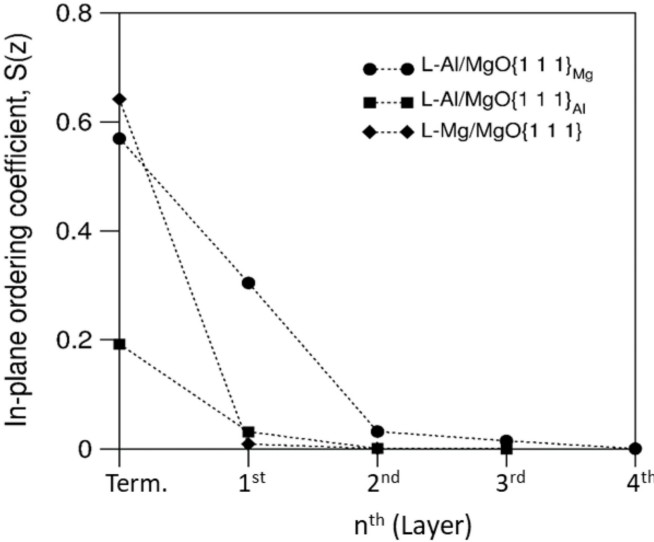

**Figure 10. The Al(*l*)/MgO{1 1 1}$_{Mg}$ interface exhibits a higher in-plane ordering coefficient.** In-plane ordering coefficients of the metallic layers adjacent to the MgO{1 1 1} substrates at the M(*l*)/MgO{1 1 1} interfaces [49,50,63].

### 3.4. General Features of the M(l)/Oxide Interfaces

Our investigation revealed a wide variety of atomic arrangements at the M(*l*)/oxide interfaces, as exampled in the previous sections. The systematic study revealed the common features of the M(*l*)/oxide interfaces, which are summarized in Table 5.

**Table 5.** Summary of the characteristics of the terminating metal layer in different M(*l*)/oxide interfaces.

| Interface | $f$(%) | M Oc.(%) | $R$(%) | $q$(e/M) | $n_{Layers}$ | $S_M(z)$ 1st $L_M$ | Prenucl. |
|---|---|---|---|---|---|---|---|
| Al(*l*)/Al{111}$_{Al}$ [10] | 0.0 | SP 100.0 | 0.0 | 0.0 | 6 | 0.50 | Strong |
| Mg(*l*)/MgO{111}$_{Mg}$ [50] | +7.9 | SP, vac. 92.0 | 4.0 | +0.60 | 3–4 | 0.01 | Weak |
| Al(*l*)/MgO{111}$_{Mg}$ [49,63] | −2.2 | SP 100 | 0.0 | +0.69 | 4 | 0.30 | Moderate-strong |
| Al(*l*)/MgO{111}$_{Al}$ [49,63] | −2.2 | MP, vac. 73.3 | 8.2 | +0.94 | 3–4 | 0.01 | Weak |
| Al(*l*)/α-Al$_2$O$_3${0001}$_{Al}$ [49,63] | +5.6 | MP, vac. 55.9 | 30.8 | +0.95 | 3 | 0.01 | Weak |
| Al(*l*)/MgAl$_2$O$_4${111}$_{Al\_2}$ [64] | +1.4 | SP, vac. 70.4 | 5.3 | +1.07 | 3–4 | 0.18 | Moderate |
| Al(*l*)/MgAl$_2$O$_4${111}$_{AlAlAl}$ [64] | +1.4 | MP, vac. 71.8 | 18.7 | +0.40 to +1.42 | 3 | 0.01 | Weak |
| Al(*l*)/MgAl$_2$O$_4${111}$_{MgAlAl}$ [64] | +1.4 | MP, vac. 75.0 | 14.7 | +0.44 to +1.17 | 3 | 0.05 | Weak |
| Al(*l*)/MgAl$_2$O$_4${111}$_{MgAlMg}$ [64] | +1.4 | MP, vac. 75.0 | 11.2 | +0.21 to +1.11 | 3 | 0.02 | Weak |
| Al(*l*)/γ-Al$_2$O$_3${111}$_{Al\_2}$ [62] | +3.3 | SP, vac. 58.1 | 11.1 | +0.80 to +1.20 | 3–4 | 0.03 | Weak |
| Al(*l*)/γ-Al$_2$O$_3${111}$_{Al\_1}$ [62] | +3.3 | MP, vac. 54.0 | 31.1 | +0.40 to +1.50 | 3 | 0.01 | Weak |

As mentioned previously, Mg and Al have different valence electrons. To keep charge balance, the different valences of Al (3+ in ionic model) and Mg (2+) cause different occupation ration ($N_{metal}/N_O$) in the bulk substrate: 100.0% for MgO, 66.7% (2/3) for Al$_2$O$_3$ and 75.0% (3/4) for MgAl$_2$O$_4$. Such charge balance influences the composition and structure of the terminating metal layer at the M(*l*)/oxide interfaces. This normalization has been applied to calculate the atomic roughness [62]. The obtained atomic roughness of the terminating metal layers is included in Table 5.

The key characteristics of the terminating M layers at the M(*l*)/oxide interfaces at thermal equilibrium are summarized in Table 5, and the key points from Table 5 are as follows:

1. Crystallographically, there is a range of lattice misfit, from moderate (1.4%) to high (7.9%);
2. Geometrically, although the terminating M layer may have a single peak in the atomic density profiles, it contains atomic vacancies and/or vertical atomic displacements and, therefore, is atomically rough.
3. Chemically, atoms in the terminating M layer are positively charged and are bonded to the substrates, becoming an integral part of the substrates;
4. The terminating metal atoms exhibit structural coupling with the metal atoms at the substrate subsurface layer, which influences the prenucleation at the interfaces.

In addition, Table 5 shows that layering at the M(*l*)/oxide interfaces ($n_{Layer} \leq 4$) is notably weaker than that of the M(*l*)/M(*s*) interface [9,10]. For instance, there is a flat and charged Mg terminating layer at the Al(*l*)/MgO{1 1 1}$_{Mg}$ interface. It has four recognizable layers, and the in-plane ordering coefficient of the first layer is 0.18 [49,63]. This suggests that the positive charging of the terminating metal atoms influences the ordering of the nearby metal atoms and thus hinders prenucleation at the interfaces.

Lattice misfit between a substrate and a solid metal has been a crucial factor affecting mainly atomic ordering at the liquid–metal/solid interfaces. Table 5 shows that the lattice misfit between MgO{1 1 1} and Mg{0 0 0 1} is significant (7.9%). At the Mg($l$)/MgO{1 1 1}$_{Mg}$ interface with a flat Mg terminating layer, there are three to four recognizable layers and 8% vacancies in the terminating layer, likely due to the lattice misfit [50]. Therefore, the weak prenucleation at Mg($l$)/MgO{1 1 1} originates from the combined effects of pronounced lattice misfit, charging, and roughness of the terminating Mg layer. Based on the data in Table 5, we analyze the relationship between atomic roughness and in-plane ordering coefficients at Al($l$)/oxide substrates, with the results presented in Figure 11.

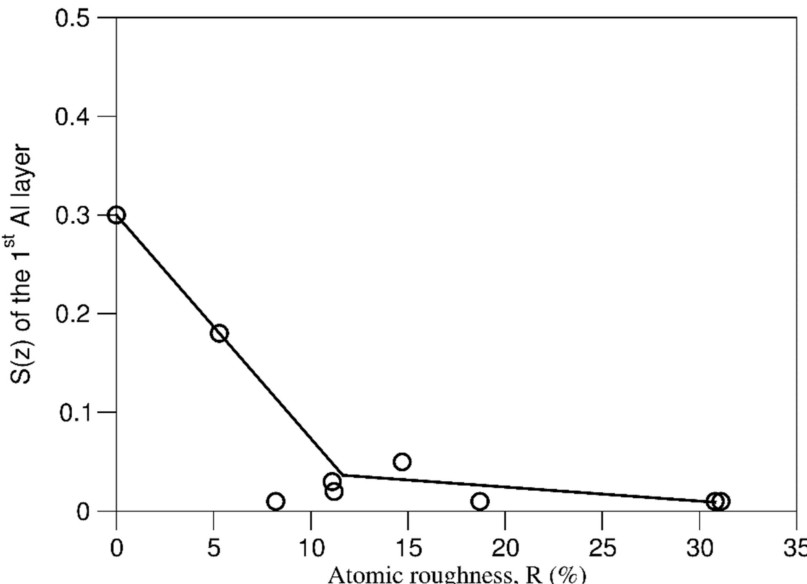

**Figure 11. Atomic ordering coefficient decreases sharply with increasing atomic roughness of the substrate surface.** Dependence of the atomic ordering coefficients, $S(z)$ on the atomic roughness, $R$ of the 1st Al layer at the Al($l$)/oxide interfaces.

Clearly, the atomic roughness of the terminating metal layer reduces the in-plane ordering in a more or less linear way with R < 10%. When R > 10%, the in-plane ordering coefficients at the M($l$)/oxide interfaces are small.

### 3.5. Potency of the Oxide Substrates and Its Role in Heterogeneous Nucleation

According to the heterogeneous nucleation theory, potency represents the intrinsic capability of a substrate to nucleate a solid phase in the liquid [6,7,15]. The prenucleation at a liquid/solid interface also relates to the intrinsic capability of the substrate to atomic template ordering in the liquid adjacent to the interface and, therefore, corresponds to the potency of the substrate for nucleation of the solid [9–11]. From Table 5, we can rank the M($l$)/oxide interfaces in terms of their capability for prenucleation (nucleation potency from high to low): Al($l$)/MgO{1 1 1}$_{Mg}$ > Al($l$)/MgAl$_2$O$_4${1 1 1}$_{Al\_2}$ > Al($l$)/$\gamma$-Al$_2$O$_3${1 1 1}$_{Al\_2}$ > Mg($l$)/MgO{1 1 1}~Al($l$)/$\alpha$-Al$_2$O$_3${0 0 0 1}~Al($l$)/$\gamma$-Al$_2$O$_3${1 1 1}$_{Al\_1}$.

The prenucleation at Al($l$)/MgO{1 1 1}$_{Mg}$ and Al($l$)/MgAl$_2$O$_4${1 1 1}$_{Al\_2}$ is more pronounced than that at the other interfaces. This suggests that the atomic roughness of the terminating metal layer hinders prenucleation. This is confirmed by the weak prenucleation at Al($l$)/$\gamma$-Al$_2$O$_3${1 1 1}$_{Al\_2}$ interface, which has a notable lattice misfit and high atomic roughness caused by vacancies.

Heterogeneous nucleation depends on the nucleation potency of substrates or the prenucleation at the M($l$)/oxide interfaces. The stronger the prenucleation is, the smaller the driving force it requires.

In order to obtain cast products of fine and uniform microstructures, attention has been paid to the potent particles as potential nucleation sites for grain refinement. The most

successful grain-refiners for Al alloys are Al-Ti-B master alloys which contain $TiB_2\{0\ 0\ 0\ 1\}$ substrates [7,55–59,77,78]. Recently, HR-TEM observations revealed that the $TiB_2\{0\ 0\ 0\ 1\}$ substrates are covered by a two-dimension compound (2DC), most likely $TiAl_3$ 2DC [59]. This 2DC enhances the potency of the $TiB_2\{0\ 0\ 0\ 1\}$ substrate for Al nucleation. In this case, the size of the $TiB_2$ particles plays a crucial role in grain initiation [5,7,16,17]. However, heterogeneous nucleation occurs on the $TiB_2$ particles of all sizes at the same temperature. When the temperature lowers to the grain initiation temperature, grain initiation starts with the large particles first and gradually occurs at the smaller ones with increasing undercooling. This grain initiation process is considered progressive [5,16], in which only a small number of large-sized particles function as grain-refinement sites.

The oxides are impotent nucleation substrates for Al and Mg alloys. Thus, they require larger nucleation undercooling, which might be lower than that of the corresponding grain initiation temperature if no other more potent particles of significance exist. Under such a situation, when the temperature reaches the nucleation temperature, the nucleation and grain initiation may occur almost simultaneously, in an explosive way [5,6,16]. On most of the substrates, grain initiation occurs. This means large fractions of particles become grain-initiation sites, and thus, the cast products may have fine and uniform microstructures.

## 4. Modification of the Terminating Metal Layers at the M(*l*)/Oxide Interfaces

Experimental observations revealed the formation of different oxide particles in alloys, e.g., coexisting of MgO and $MgAl_2O_4$ spinel particles in Al-Mg alloys and $\alpha$- and $\gamma$-$Al_2O_3$ particles in Al-rich alloys. These oxide particles exhibit a wide variety of nucleation potency (Table 5). Moreover, as shown in Figure 1 and Table 5, one type of oxide particle, e.g., $\gamma$-$Al_2O_3$ particles, may have terminating surfaces of different nucleation potency [62]. In liquid, these oxide particles compete for heterogeneous nucleation, and consequently, only some oxide particles of higher potency may participate in the nucleation process [5,16].

Wang et al. investigated the cast samples of Mg-0.5Ca alloy and observed segregation of impurities, including Ca on the MgO{1 1 1} substrates [79]. Ma et al. studied the effects of Cu segregation at the sapphire substrate on the nucleation of Al using an AIMD simulation technique [80]. Recently, Wang et al. observed the segregation of La and Y atoms on Al(*l*)/$\gamma$-$Al_2O_3\{1\ 1\ 1\}$ substrates using the HR-TEM techniques [81]. To have a better knowledge of the segregation of the $nd^1$ elements, Sc, Y, and La atoms at the oxide interfaces, we performed systematic AIMD simulations [82]. Figure 12 shows the snapshots (Figure 12a,c) and atomic density profiles (Figure 12b) of the equilibrated Al(*l*)/$\gamma$-$Al_2O_3\{1\ 1\ 1\}$ interfaces with the segregation of La atoms, as an example.

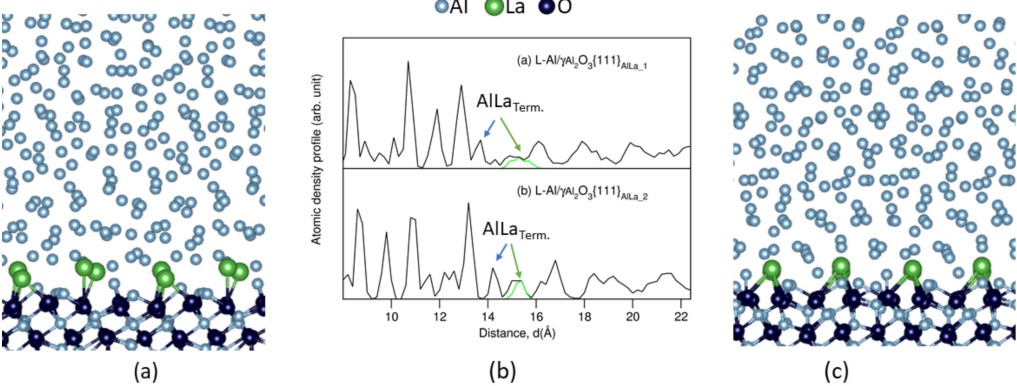

**Figure 12. Segregation of La atoms at the Al(*l*)/$\gamma$-$Al_2O_3\{1\ 1\ 1\}$ induces atomic roughness at the oxide substrates.** Snapshots of Al(*l*)/$\gamma$-$Al_2O_3\{1\ 1\ 1\}_{(AlLa)\_1}$ (**a**) and Al(*l*)/$\gamma$-$Al_2O_3\{1\ 1\ 1\}_{(AlLa)\_2}$ (**c**) systems and their atomic density profiles (**b**) equilibrated at 1000 K [82].

The AIMD simulations revealed the formation of a 2DC layer of $Al_2La$ at both Al(*l*)/$\gamma$-$Al_2O_3\{1\ 1\ 1\}_{Al\_1}$ and Al(*l*)/$\gamma$-$Al_2O_3\{1\ 1\ 1\}_{Al\_2}$ interfaces. The terminating Al/La layers are composed of multiple peaks and thus become atomically rougher (Figure 12b) as compared

with those Al(*l*)/γ-Al$_2$O$_3${1 1 1} interfaces without segregation. Such surface roughening deteriorates the terminating capability of the Al(*l*)/γ-Al$_2$O$_3${1 1 1}$_{Al\_2}$ interface. Consequently, the nucleation potency of both γ-Al$_2$O$_3${1 1 1}$_{(Al/La)\_1}$ and γ-Al$_2$O$_3${1 1 1}$_{(Al/La)\_2}$ substrates is reduced.

## 5. Summary and Perspective

We presented an overview of the recent advances in ab initio studies on the prenucleation at the interfaces between oxide substrates and liquid metals. There is an ordered metal layer terminating the substrates in liquid metals. This newly formed terminating metal layer strongly affects prenucleation at the interfaces. The terminating metal atoms are positively charged and chemically bonded to the substrates, being an integral part of the substrates. Topologically, the layers may be atomically rough. The prenucleation at these interfaces provides a basis for thermodynamic analysis and modeling [83,84].

Charging of the terminating metal atoms weakens the prenucleation at the interfaces. This is new in interfacial interaction chemistry. Atomic roughness at oxide surfaces deteriorates both layering and in-plane-ordering, which can be utilized to promote explosive grain initiation. The obtained information is helpful in obtaining insight into the prenucleation at the various interfaces between liquid metals, including iron, titanium, etc., and solid substrates, such as oxides and dipolar AlN, SiC, etc.

This study is helpful for understanding solidification and for the manipulation of solidification processes. Based on the present study, it is possible to manipulate the atomic arrangement at the substrate surfaces via, e.g., segregating foreign atoms on the substrate surfaces. This is exampled by the La atoms segregation at the Al(*l*)/γ-Al$_2$O$_3${1 1 1} interfaces.

**Author Contributions:** Conceptualization, C.F. and Z.F.; methodology, C.F.; software, C.F.; validation, C.F.; formal analysis, C.F.; investigation, C.F.; resources, C.F.; data curation, C.F.; writing—original draft preparation, C.F.; writing—review and editing, C.F. and Z.F.; visualization, C.F.; supervision, Z.F.; project administration, Z.F.; funding acquisition, Z.F. All authors have read and agreed to the published version of the manuscript.

**Funding:** Financial support from EPSRC (UK) under grant number EP/N007638/1 is gratefully acknowledged.

**Acknowledgments:** We thank Yun Wang and Hua Men for their useful comments.

**Conflicts of Interest:** The authors declare no conflict of interest.

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
