# Peer review of "Ab Initio Molecular Dynamics Investigation of Prenucleation at Liquid–Metal/Oxide Interfaces: An Overview"

_metals, doi:10.3390/met12101618_

Round 1

Reviewer 1 Report

Ab initio molecular dynamics (AIMD) techniques is a resource-intensive procedure. The authors performed a review based on 16 publications. The authors study only the prenucle ation at the interfaces between oxide substrates and liquid metals. This study is useful for understanding solidification and controlling solidification processes. This approach can be used in related areas (2D heterostructures). Therefore, it may be of interest to a wider readership.

Author Response

We thank the Reviewer 1 for the carful reading and the constrictive comments and suggestions. We here respond their comments and suggestions. The comments are labelled by C and the responses by Re. 

Reviewer 1: 

C: Ab initio molecular dynamics (AIMD) techniques is a resource-intensive procedure. The authors performed a review based on 16 publications. The authors study only the prenucleation at the interfaces between oxide substrates and liquid metals. This study is useful for understanding solidification and controlling solidification processes. This approach can be used in related areas (2D heterostructures). Therefore, it may be of interest to a wider readership. 

Re: We thank Reviewer 1 for the careful reading and understanding of the present overview on AIMD simulation of the Al(l)/oxide interfaces. We agree with Reviewer 1 that this approach can be used in related areas and might be interesting to a wider readership. 

Reviewer 2 Report

The Author attempt to provide a review of ab initio studies of prenucleation at oxide and liquid metal. The effect of terminating metal atoms(its lattice mismatch, charge effects) and surface roughness were discussed.

The paper is well written and provides the needed details to support its conclusion. But the papers are overstressing on AL/Al2O3 interfaces which is not consistent with the general title of liquid metal /oxide interface. If author can provide more references to other types of interfaces it would be better.

Of the 84 references about 30 papers are self-cited work though the work is properly cited this effort seems to be a review of the author's research rather than the body of knowledge. Adding more types of interfaces would help resolve this issue.

Author Response

We thank the Reviewer 2 for the carful reading and the constrictive comments and suggestions. We here respond their comments and suggestions. The comments are labelled by C and the responses by Re in the attached file

Reviewer 3 Report

In order to understand the nature of casting techniques, which remain the essential material production technique, it is vital to comprehend prenucleation in truly molten metal. In this respect, the study will significantly contribute to the literature. In the study, quite a lot of references have been examined in detail and analysed. The results have been presented by interpreting in the nature of a review. For this reason, it is recommended that the study be published with minor revisions.

1. It would be more appropriate not to use abbreviations in the title, so molecular dynamics should be used instead of MD.

2. In the first sentence of the introduction, the information “The densities of the light metals, Mg and Al are about one-fifth and one-third of that of iron, respectively [1, 2]” is given. This information is general literature information and is known to everyone. Therefore, references are unnecessary. References should be removed.

3. Page 2: Oxide particles, including magnesia, (α- and γ-) alumina and spinel form inevitably in liquid alloys during melting handing and casting [18-28]. The reason for giving 11 references for one sentence was not understood. There is a lot of unnecessary use of references throughout the study.

4. While explaining Figure 1, attention should be paid to the order in the text. For example, Figure 1e was given after Figure 1a.

5. There is a lot of unnecessary use of references throughout the study.

Author Response

We thank the Reviewer 3 for the carful reading and the constrictive comments and suggestions. We here respond their comments and suggestions. The comments are labelled by C and the responses by Re in the attached file

Round 2

Reviewer 2 Report

The response by the author is satisfactory hence I support the publication of this paper in its current form